# The Impact of Health Education on the Quality of Life of Patients Hospitalized in Forensic Psychiatry Wards

**DOI:** 10.3390/ijerph20054533

**Published:** 2023-03-03

**Authors:** Joanna Fojcik, Michał Górski, Agnieszka Borowska, Marek Krzystanek

**Affiliations:** 1Doctoral School, Faculty of Health Sciences in Katowice, Medical University of Silesia, 40-055 Katowice, Poland; 2Doctoral School, Faculty of Health Sciences in Bytom, Medical University of Silesia, 41-902 Bytom, Poland; 3Department of Psychiatric Rehabilitation, Leszek Giec Upper-Silesian Medical Centre, Medical University of Silesia, 40-055 Katowice, Poland; 4Department and Clinic of Psychiatric Rehabilitation in Katowice, Faculty of Medical Sciences in Katowice, Medical University of Silesia, 40-055 Katowice, Poland

**Keywords:** health education, forensic psychiatry, schizophrenia

## Abstract

Purpose: An original health education program, developed for a group of patients of forensic psychiatry wards, was the basis for conducting a study on the impact of educational influences on the quality of life of patients long-term isolated from their natural environment. The main aim of the study was to answer the question: Does health education affect the quality of life of patients in forensic psychiatry wards and is educational activity effective? Methods: The study was conducted at the State Hospital for Mental and Nervous Diseases in Rybnik, Poland, in the forensic psychiatry wards, and lasted from December 2019 to May 2020. During the study, patients gained knowledge in the field of broadly understood health education. The study group consisted of 67 men, aged 22–73, diagnosed with schizophrenia. The method of double measurements (before and after the health education cycle) was applied, using the WHOQOL-BREF scale of quality of life and the first author’s questionnaire of patients’ knowledge, from the educational program used. Results: Health education does not significantly affect the overall quality of life of patients staying in forensic psychiatry wards, but it does affect their somatic condition. The proprietary health education program is effective because the patients’ knowledge has significantly improved. Conclusions: The quality of life of interned patients with schizophrenia is not significantly related to educational activities, however, psychiatric rehabilitation through educational activities effectively increases the level of patients’ knowledge.

## 1. Introduction

Research on the quality of life began in the early 1960s and 1970s, and in recent years this issue has received more structured interest from researchers in various fields of science. The concept of quality of life is ambiguous, multidimensional, and multidisciplinary, and reflects many aspects of human functioning. To a large extent, it is a subjective value and depends on a person’s mental state, personality, preferences, and value system.

The subjective assessment of a patient’s quality of life is still relatively little known, and scientific work on it has been scarce. In psychiatry, this situation is a bit more complicated, because the subjective factor in the assessment of the patient’s mental state is of great diagnostic and prognostic importance. Therefore, systematic studies of the assessment of the quality of life of patients with mental disorders were undertaken, with some reluctance, and were delayed in relation to the studies of the quality of life of somatic patients [1].

Reflections on the quality of life of patients with schizophrenia bring the question of whether a schizophrenic patient is able to accurately assess their quality of life, due to the lack of insight and cognitive deficits often associated with the disease. Scientists believe that patients with schizophrenia are aware of their own social deficits and the information obtained from them is useful in the process of diagnosis and treatment. Regardless of whether the subjective dimension is consistent with the patient’s objective situation or not, it remains important in the holistic assessment of the patient’s mental state [2,3,4].

Patients treated in forensic psychiatry wards are an extremely difficult group of patients. These are not only mentally ill people but also perpetrators of acts prohibited by law. These patients usually have a long criminal history, suffer from severe mental disorders, are often drug resistant, addicted to psychoactive substances, and very often have severe personality disorders co-occurring. Working with such a patient is a long-term and multidimensional process and is not based solely on providing services resulting from the presence of psychopathological symptoms.

For these reasons, in addition to pharmacological and psychotherapeutic treatment, patients are provided with a wide range of sociotherapeutic, rehabilitation, and resocialization interactions, which are a set of interactions that are designed to lead to the patient’s proper functioning in society, in accordance with the accepted social and legal order. These interactions are a special form of a multidimensional approach that includes elements of education (i.e., education and upbringing), care, and therapy. Return to society, readaptation, coping with the disease, and restoring hope for a satisfying life are the main goals of education and upbringing in forensic psychiatry wards, and the patient’s participation in this process is a factor building the patient’s co-responsibility for their own health [5].

Scientific research clearly shows that pharmacological treatment, in combination with psychosocial interactions, is an important element of therapeutic programs aimed at helping people with schizophrenia recover [6,7]. The function of health educators among this specific group of patients is often taken on by nurses who, in addition to standard nursing procedures, conduct psychiatric rehabilitation activities closely related to, among other things, the health education of patients. The nurse of the forensic psychiatry ward is the person closest to the patient, in light of this they are an unquestionable source of information about the patient’s changing physical and mental condition. The knowledge about the patient obtained by nurses is a reliable foundation for both treatment and rehabilitation, because patients of forensic psychiatry wards stay there long after their mental illness has stopped being a leading problem.

The health education program for mentally ill offenders, developed by the first author, was the starting point for examining the impact of educational programs on the quality of life of patients long-term isolated in a forensic psychiatry ward. So far, no such studies have been conducted. The obtained results may become a premise for standardizing the work of nurses and developing a model of patient care in the forensic psychiatry ward, as well as be used to develop therapeutic and rehabilitation programs for patients with mental disorders, where the element of education will be an important part of the rehabilitation process.

## 2. Material and Methods

The study was conducted at the State Hospital for Mental and Nervous Diseases in Rybnik, Poland, in five units of forensic psychiatry. The study group consisted of 67 men, aged 22–73, with a diagnosis of schizophrenia. The study lasted for 6 months, from December 2019 to May 2020, during which patients gained knowledge and social competences during lectures in the field of broadly understood health education. The reference group in the study was a group of 48 patients interned in a forensic psychiatry ward, for whom no health educational activities were conducted. The statistical analysis indicated that the study and reference groups did not differ in a statistically significant way (Table 1).

The health education program was structured, individualized, and adapted to the educational needs of patients hospitalized in forensic psychiatry wards. The educational process in which they participated was intended to increase knowledge about mental illness, including its causes, symptoms, dynamics, and treatment options, but also to develop social skills. The assessment of the effect of health education was carried out with a knowledge test, performed twice, before and after the educational cycle. The knowledge test carried out before the series of educational lectures was intended to assess the initial level of patients’ knowledge of topics that would appear in the series of lectures. The health education program consisted of 40 topics related to social life, mental health, healthy lifestyle, and functioning of the patient in the forensic psychiatry ward. Patients participating in the study attended educational group lectures twice a week, for a period of 6 months. After the completion of the medical education cycle, the same knowledge test was performed again.

## 3. Results

A total of 115 patients of forensic psychiatric wards, diagnosed with schizophrenia, participated in the study, of which, data on 101 patients were obtained, 61 patients in the study group and 40 patients in the reference group. Out of the initial number of 67 patients in the study group, 61 patients completed the six-month health education cycle after the first assessment with the knowledge test. Of the 48 patients in the reference group in the first measurement, 40 had a second measurement after 6 months. A total of 14 patients from both groups did not complete the study, due to discharge from the hospital, refusal to complete research questionnaires, or transfer of the patient to a non-forensic psychiatry ward.

### 3.1. WHO Quality of Life (WHOQOL-BREF) Scale

Table 2 shows how patients answered the question related to the overall assessment of their quality of life. In the first measurement, the quality of life was negatively assessed by 7.5% of the study group and 6.3% of the reference group (answer “very bad” or “bad”). The quality of life was positively assessed by 58.3% of the study group and 48% of the reference group (answer “good” or “very good”). In the study group, 34.3% of patients and in the reference group, 45.8%, did not specify the assessment of the quality of life, marking the answer “neither good nor bad”.

In the second assessment, the percentage values of quality of life in the study group increased, both in relation to the positive assessment, by 2.4% (60.7% of the group), and negative, by 4% (11.5% of the group). However, the number of people who could not define their quality of life decreased, to 27.9% of the group. In the reference group, in the second measurement, the percentage values of the negative assessment increased by 6.2% (12.5% of the group), and the unspecified assessment, by 4.2% (50% of the group), while the positive assessment of the quality of life decreased by 10.5% (37.5% of the group).

The Wilcoxon paired-order test within each of the obtained groups did not show statistically significant individual changes in the assessment of the quality of life. There was also no statistically significant difference in the distribution of individual changes between the study and reference groups. Detailed results are presented in Table 2.

Descriptive statistics of individual domains of the WHOQOL-BREF scale in the study group and the control group for the 1st measurement are presented in Table 3, and for the 2nd measurement in Table 4.

For each of the groups of analyzed results, no statistically significant difference in their distributions compared to the theoretical normal distribution was found. Statistical analysis of mean differences of individual domains from the study and control groups, performed with the Student’s *t*-test for unrelated samples, showed statistical significance (*p* < 0.001101) only for the 3rd domain (social status). The higher values of this domain were observed in the study group.

No statistically significant difference was found in the distributions of any of the groups compared to the theoretical normal distribution. Statistical analysis of the differences in the means of individual domains from the study group and the control group, using the Student’s *t*-test for unrelated samples, showed a statistically significant result in the 3rd domain (social status).

Table 5 presents the results of descriptive statistics of changes in domain values between the first and second measurement within each group, and the result of the test of significance of changes in the group. In the second measurement, an increase of 3.7 points in the mean value in domain 1, and a decrease of 5.5 points in the mean value in domain 3, were observed in the study group. In none of the analyses was a significant result obtained at the significance level of *p* = 0.05. The significance levels obtained in the study group for domain 1 (*p* = 0.09) and domain 3 (*p* = 0.06) indicate the occurrence of a statistical trend related to the conducted medical education cycle, regarding the increase in patients’ assessment of their somatic condition (domain 1) and decrease in their assessment of social status (domain 3).

To determine whether the domain values are correlated with the qualitative factors given in the characteristics of the groups, analysis of covariance (ANCOVA) or variance (ANOVA) was performed. The results of men who had two measurements were included in the analyses. The qualitative factors analyzed are shown in Table 6.

Table 6 also shows the significance level for individual effects contributing to the dependent variable (WHOQO-BREF scale domain value), by the analyzed pairs of variables (ANCOVA analysis) or only the categorical variable (ANOVA analysis). For the remaining domains of the relationship, the quantitative variable (age of life) for the second measurement was not analyzed, due to the lack of correlation shown in other analyses.

In the study group, in all ANCOVA analyses, a statistically significant result was obtained for the results of the first measurement, indicating a relationship between the values of the analyzed domains and the current age of life. The ANCOVA analysis in the study group showed that the form of professional activity of patients, has a significant impact on the values of domain 1 (somatic condition) and domain 2 (psychological condition), both in the results of the first and second assessments. Patients who were on disability pension assessed their somatic condition (domain 1) and psychological condition (domain 2) the worst, in comparison to other respondents.

The ANCOVA analysis showed, in the first evaluation, that the mean domain 1 values are related to the severity of the clinical condition expressed on the CGI-S scale. A significant impact of the level of mental disturbances on domain 1 values was found only before the start of the training cycle. At the end of the cycle, such statistical significance was not demonstrated.

In the performed ANCOVA covariance analysis, a significant result was also obtained for the impact of the number of correct answers in the knowledge test on the value of the D1 domain (somatic condition) in the first assessment. The results indicate lower average domain 1 values for fewer correct answers, before the start of the education cycle (*p* = 0.02). Such results indicate that patients with a lower level of medical education, expressed in the number of correct answers, assess their quality of life, described by their somatic condition, worse. Along with the increase in the level of knowledge in the field of health education, the self-assessment of the quality of life in the domain of somatic condition increases. For the results of the second assessment, after the end of the six-month health education cycle, the mean domain values did not differ significantly (*p* = 0.89).

Table 7 shows the interdependence of changes in the number of correct answers to the questions of the knowledge test, with changes in the values of individual domains of the WHOQO-BREF quality of life scale, among all patients in the study group who underwent two measurements, and those who in the first measurement answered correctly to less than 34 knowledge test questions. An increase in the number of correct answers in the entire group of patients, after the completion of the health education training cycle, was found in 39 men (63.9% of the group). Expanding knowledge in the field of health education, identified with the number of correct answers, occurred statistically significantly more often than in 50% of the entire group (*p* = 0.01). The value of 50% is defined in statistics as a probable occurrence.

Of these 39 patients, 20 (32.8% of the group) also had an increase in domain 1, 19 (31.3% of the group) an increase in domain 2, 12 (19.7% of the group) an increase in domain 3, and 16 (26.2% of the group) an increase in domain 4. In the group of 30 patients with less than 34 correct answers to the knowledge test in the first assessment, an increase in the number of correct answers in the second assessment was found in 19 men, which is statistically significantly for greater than 50% of the group (*p* = 0.05). Of these 19 patients, 11 (36.7% of the group) also had an increase in domains 1 and 2, 6 (20.0% of the group) an increase in domain 3, and 8 (26.7% of the group) an increase in domain 4.

Table 7 also includes the calculated OR odds ratios (with 95% confidence interval) of the increase in the domain value, with an increase in the number of correct answers compared to those in which the increase in the number of correct responses did not occur. For the entire study group, the estimated OR values ranged from 1.00 for domain 4, to 2.53 for domain 2. OR values above 1.00 indicate a greater chance of an increase in the domain value with an increase in the number of correct answers. None of these values turned out to be statistically significant, due to the wide confidence intervals.

The greatest chance of domain growth, associated with an increase in the number of correct answers, was found for domain 2 (psychological state). Here the chance is 2.53 times greater than the chance of domain growth with no increase in the number of correct answers. Although the value was not statistically significant, the calculated significance level of *p* = 0.09, may indicate a presence of statistical tendency. For domain 3, the odds ratio was OR = 1.51, and for domain 1, OR = 1.26. The obtained values indicate that the chance of an increase in the self-assessment of the quality of life in these domains is greater than without improving the state of knowledge in the field of medical education.

Analyzing the results among patients who answered less than 34 questions correctly in the first assessment, a statistically significant result was obtained for the odds ratio OR = 6.19, domain 2. The chance of an increase in self-assessment of mental health (domain 2) among patients with an initial low level of medical knowledge after the cycle of educational training, is over 6 times greater than the chance of domain growth with no improvement in knowledge.

The results of Table 7, based on retrospective data, allow us to conclude that conducting a series of lectures in the field of medical education probably increases the level of medical knowledge in patients, which may affect the change in self-assessment of health, especially in the psychological sphere (domain 2).

### 3.2. Health Education Knowledge Test

The general knowledge of men in the field of health education was assessed as the total number of correct answers to all questions of the test. The parameters of the descriptive statistics of the number of correct answers in the two assessments, in the study and reference groups, are presented in Table 8. In the study group, the number of correct answers of patients in the first assessment ranged from 19 to 40, and in the second assessment from 17 to 40. The average number of correct answers in the first assessment in the study group was 32.9, and in the reference group it was 34.3. The variations of the results described by the standard deviation of these measurements were 4.0 and 4.6, respectively. In the reference group, the range of changes in the number of correct answers in the first measurement was the same as in the study group, and in the second measurement it ranged from 22 to 39. The average results were 31.2 and 30.5, respectively, and the standard deviations were 5.1 and 4.3. Percentile values in individual groups of results are presented in Table 8.

The distribution of the analyzed numbers of correct answers to the questions of the knowledge test deviated in three cases from the theoretical normal distribution (Kolmogorow–Smirnov test). The Mann-Whitney U test did not show a statistically significant difference in the results between the groups for the first assessment, while for the second assessment statistical significance was obtained. In the Wilcoxon test, comparing the results from the two assessments, a statistically significant difference was shown only for the study group. The results of these analyses indicate that, after the health education cycle, the medical knowledge of patients in the study group significantly improved. 

### 3.3. Summary of Results

The conducted study, the main purpose of which was to determine the impact of health education on the quality of life of patients of forensic psychiatry wards, indicates that, in terms of assessing the quality of life, educational activities carried out, based on the health education program, in the group of patients did not have a significant impact on their overall quality of life. However, they showed a steady upward trend of scoring in the assessment of the quality of life, after the cycle of health education, in the somatic domain, but not in a significant way. The age of the patients’ life had a significant impact on their assessment of their quality of life in the initial phase of the study, i.e., before the health education cycle, patients assessed their quality of life worse. In the field of health education, the program implemented among patients of forensic psychiatry wards turned out to be effective and significantly improved the patients’ knowledge.

## 4. Discussion

Studying the quality of life of patients in forensic psychiatry wards is a big challenge. It should be noted that these patients remain in continuous isolation for many years and it is difficult to talk about a good quality of life of interned patients. A long stay of a patient in the ward does not have a positive effect on their well-being. It is a source of internal conflicts, a sense of injustice and frustration. Forced isolation for such a long time means that the assessment of the quality of life of patients is influenced by many factors, not necessarily positive, such as, for example, the decision to extend forced hospitalization. In addition, unpublished research by the authors also shows that patients of forensic psychiatry wards do not have adequate support from relatives, often they feel left alone, which deepens their feeling of isolation. These are just some of the factors that accompany patients and may have a negative impact on the assessment of their quality of life. It should also be remembered that a patient in a psychiatric ward is under constant observation and their health, behavior, and participation in treatment and therapy are monitored.

For those reasons patients of forensic psychiatry wards are therefore unable to maintain their physical and psychological autonomy, they stay in an artificially created environment for many years, which causes numerous limitations in everyday functioning, which certainly affects the quality of their lives. The patient of the forensic psychiatry ward is a compulsorily hospitalized patient who, above all else, wants to regain their freedom, therefore their answers to all kinds of tests should be treated with great caution, because they may want to present themselves in a way that is favorable to them.

The study of the relationship between the impact of health education on the quality of life of patients was preceded by asking patients about their general assessment of their quality of life, to be able to globally determine its level in this particular group of patients. Both before and after the health education cycle, patients very similarly assessed their quality of life, so it can be concluded that education has no significant impact on their assessment of their quality of life. The obtained results are similar to the results of other studies, which have indicated that, for example, an education and an increase in the level of knowledge about schizophrenia do not translate into an increase in the subjective assessment of the overall quality of life [8,9]. The obtained results also confirm the results of another study, which showed that greater criticism of the disease obtained through educational activities, and thus increased awareness of the disease and its consequences, is associated with a lower assessment of patients’ quality of life [10].

Interestingly, a detailed analysis of the results of this study showed that most patients assess their overall quality of life at a satisfactory level, and only a small group of patients present a negative opinion. Although the overall assessment of the quality of life of the surveyed patients turned out to be satisfactory, other studies clearly indicate that people with mental disorders assess their quality of life worse than healthy people [11,12,13,14]. Schizophrenia is a severe mental disorder characterized by positive and negative symptoms and cognitive deficits. Compared to healthy individuals, patients with schizophrenia are at greater risk for comorbid physical illnesses, cognitive and occupational impairments, frequent hospitalizations, high medical costs, and increased risk of suicide and mortality, all of which come with a heavy personal and family burden that undoubtedly impacts their quality of life. It is therefore possible that patients deprived of liberty, free from factors unfavorable to mental health, including the stigma of the disease, can assess the quality of their life adequately to the conditions in which they currently find themselves, and it is possible that, if the internment lasts a very long time and the patient is deprived of stressors that people with schizophrenia experience in free conditions, their assessment of their quality of life is definitely better.

Thanks to the WHOQOL-BREF standardized quality of life scale used in this study, it was possible to assess the quality of life of patients in forensic psychiatry wards in a detailed way, focusing on the assessment in several domains: somatic condition, psychological state, social status, and environment. Based on the conducted research, it can be concluded that the participation of patients in the health education cycle changes the values of individual domains. For the assessment of the somatic condition of patients, this change turns out to be beneficial—education improves their somatic well-being, probably also by increasing self-awareness, which, however, also causes a decrease in their social self-esteem. Similarly, other scientific studies indicate that some sociodemographic and clinical characteristics affect the quality of life of patients with schizophrenia. These results deepen the knowledge about these characteristics and should be considered in the clinical assessment of the patient and in planning appropriate and effective strategies for their psychosocial rehabilitation [11,15,16].

In the presented results concerning the analysis of pairs of variables, it was found that the assessment of the quality of life in the somatic domain is significantly affected by such factors as professional activity, severity of disease symptoms, and the number of correct answers in the knowledge test. It follows that patients with a lower level of medical education assess their quality of life related to their somatic condition worse. The situation changed after the medical education cycle, when the self-assessment of the quality of life in the domain of somatic condition improved, along with the increase in the level of knowledge in the field of health education. The obtained results indicate that the participation of patients in the cycle of health education has a positive effect on their quality of life in the somatic sphere. The conducted analyses show that the general assessment of the quality of life did not show a significant correlation with the education process, however, the somatic component of the scale of quality of life changed positively.

Since educational impacts affect the quality of life of patients in the somatic aspect, this information is not only important from the point of view of the care of a patient staying in a forensic psychiatry ward. This conclusion can be applied to patients with schizophrenia in general and may be useful for therapists who want to introduce the process of patient health education to their work with patients diagnosed with schizophrenia. The obtained results confirm the conclusions of some studies of patients with schizophrenia, which have showed that educational activities in the field of promoting a healthy lifestyle are significantly related to the results of the WHOQOL-BREF quality of life questionnaire [17]. The analysis of the odds ratios in the conducted study, also showed that conducting a series of educational lectures in the field of health education is likely to increase the level of medical knowledge in patients, which may change the self-assessment of their health, especially in the psychological domain.

The study also showed that educational activities are effective in knowledge improvement. A set of 40 issues necessary to conduct educational lectures in the field of broadly understood health education, preceded by a knowledge test, estimated the initial level of knowledge of patients, both from the study group and the reference group. The analysis of the results showed that after the health education cycle, the medical knowledge of patients in the study group significantly improved, which was not shown in the reference group. This proves the effectiveness of educational activities in this group of patients. Increasing the medical knowledge of patients may have pro-health implications for them. There are many publications on the effectiveness of this form of therapy and the methods of its conduct [18,19,20]. The relationship of participation in psychoeducation with shorter hospitalization time, fewer relapses, improved health and psychosocial functioning of patients, their better cooperation, and greater knowledge about the disease has been previously demonstrated [21,22,23,24,25].

These observations have important clinical implications. The main therapeutic goal in forensic psychiatry wards is to prepare patients for life in freedom, in accordance with applicable law and social norms, and in such a way as to minimize the risk of re-committing a criminal act. Undoubtedly, all educational activities undertaken by patients are key tools to achieving this goal. The results of this study confirm the possibility of improving their condition through the health educational program. The fact that these interactions improve the patient’s knowledge, and thus contribute to a greater awareness of the patient’s life with a mental illness and all its consequences, gives hope for improving social functioning, and thus a chance to live in accordance with its principles.

Providing educational information and involving patients in treatment has become an important and effective element of psychiatric care, which has been confirmed by numerous scientific studies [26,27,28,29,30]. Each psychosocial impact, as well as rehabilitation, neutralizes the causes of patients’ withdrawal from social life and teaches them to return to a situation in which they could function properly in their environment, which in relation to the group of patients of forensic psychiatry wards is an extremely important clue in the process of therapy and treatment [17,31].

Scientific research clearly shows that pharmacological treatment, combined with psychosocial interactions, is an important element of therapeutic programs aimed at helping people with schizophrenia recover [6,7]. Since the main purpose of the patient’s stay in a forensic psychiatry ward is to prepare them for life in freedom, in accordance with the applicable social norms, the inclusion of non-pharmacological forms of treatment becomes not only a method but also a somewhat ethical obligation. It is not only an addition to pharmacological treatment, but an integral part.

## 5. Conclusions

The global quality of life of interned patients with schizophrenia is not significantly related to educational activities, however, sub-domain analysis indicates that health education improves their somatic well-being. Psychiatric rehabilitation through educational activities effectively increases the level of patients’ knowledge.

## Figures and Tables

**Table 1 ijerph-20-04533-t001:** Comparison of differences between the studied groups.

Variable	Total n = 115; 100%	Study Group n = 67; 100%	Control Group n = 48; 100%	*p*-Value *
Age (years)				
<25	5; 4.4%	2; 2.9%	3; 6.2%	0.31 *
25–35	29; 25.2%	19; 28.4%	10; 20.8%	
36–45	29; 25.2%	19; 28.4%	10; 20.8%	
46–55	23; 20.0%	14; 20.9%	9; 18.8%	
56–65	23; 20.0%	10; 14.9%	13; 27.2%	
>65	6; 5.2%	3; 4.5%	3; 6.2%	
Marital status				
Single	93; 80.9%	56; 83.6%	37; 77.1%	0.13 **
Married	6; 5.2%	1; 1.5%	5; 10.4%	
Divorced	16; 13.9%	10; 14.9%	6; 12.5%	
Housing status				
I live alone	50; 43.5%	25; 37.3%	25; 52.1%	0.23 **
I live with my family	56; 48.7%	35; 52.2%	21; 43.8%	
Institutional care	9; 7.8%	7; 10.4%	2; 4.2%	
Education				
None	3; 2.6%	2; 3.0%	1; 2.1%	0.07 **
Primary education	23; 20.0%	13; 19.4%	10; 20.8%	
Lower secondary school	8; 7.0%	3; 4.5%	5; 10.4%	
Professional	47; 40.9%	22; 32.8%	25; 52.1%	
Secondary education	28; 24.3%	23; 34.3%	5; 10.4%	
Higher	6; 5.2%	4; 6.0%	2; 4.2%	
Length of stay in the ward				
Up to 1 year	28; 24.3%	21; 31.3%	7; 14.6%	0.31 **
1–3 years	46; 40.0%	25; 37.3%	21; 43.8%	
4–5 years	20; 17.4%	7; 10.4%	13; 27.1%	
6–9 years	10; 8.7%	6; 9.0%	4; 8.3%	
>9 years	11; 9.6%	8; 11.9%	3; 6.3%	
Antipsychotics used				
Classic	29; 25.2%	13; 19.4%	16; 33.3%	0.13 **
Atypical	86; 74.8%	54; 80.6%	32; 66.7%	

* Student’s *t*-test, ** Yates-corrected chi^2^ test.

**Table 2 ijerph-20-04533-t002:** Distributions of responses to question 1, “What is your quality of life”, of the WHOQOL-BREF scale survey questionnaire, among patients of the study group, and references for the 1st and 2nd measurements.

Answer	1st Measurement		2nd Measurement	
	Study Group (n = 67; 100%)	Reference Group (n = 48; 100%)	Study Group (n = 61; 100%)	Reference Group (n = 40; 100%)
Very bad */	2 (3.0%)	1 (2.1%)	0	1 (2.5%)
Bad */	3 (4.5%)	1 (4.2%)	7 (11.5%)	4 (10.0%)
Neither good nor bad	23 (34.3%)	22 (45.8%)	17 (27.9%)	20 (50.0%)
Good	32 (47.8%)	21 (43.8%)	27 (44.3%)	12 (30.0%)
Very good	7 (10.5%)	2 (4.2%)	10 (16.4%)	3 (7.5%)
Chi^2^ test with Yates correction of comparisons of distributions between groups	NS (*p* = 0.46)		NS (*p* = 0.13)	
Wilcoxon paired-order test for group changes	Study group: NS (*p* = 0.46) Reference group: NS (*p* = 0.50)			

*/—for statistical analysis, categories combined.

**Table 3 ijerph-20-04533-t003:** Descriptive statistics of the domains (standardized at 0–100 points) of the WHOQOL-BREF scale for 1st measurement of the study and reference groups.

**1st Assessment**								
Statistical Parameter	D1. Somatic Condition		D2. Psychological State		D3. Social Status		D4. Environment	
	Study Group	Reference Group	Study Group	Reference Group	Study Group	Reference Group	Study Group	Reference Group
Numbers	67	48	67	48	67	48	67	48
Average	62.8	60.4	67.2	64.3	63.4	47.9	62.1	56.7
Standard deviation	16.8	17.2	16.4	19.2	20.0	20.9	15.1	18.0
Minimum	21	0	37	0	17	0	22	0
25th percentile (bottom quartile)	54	49	54	53	50	33	53	47
50th percentile (median)	64	61	67	67	67	50	59	56
75th percentile (upper quartile)	75	71	79	79	75	60	75	69
Maximum	96	89	100	100	100	92	100	94
Test of normality of distribution Kolmogorov–Smirnov	NS (*p* > 0.20)	NS (*p* > 0.20)	NS (*p* > 0.20)	NS (*p* > 0.20)	NS (*p* > 0.10)	NS (*p* > 0.20)	NS (*p* > 0.20)	NS (*p* > 0.20)
Student’s *t*-test for unrelated samples	NS (*p* = 0.45)		NS (*p* = 0.39)		*p* < 0.0011		NS (*p* = 0.08)	

**Table 4 ijerph-20-04533-t004:** Descriptive statistics of the domains (standardized at 0–100 points) of the WHOQOL-BREF scale for the 2nd measurement in the study and reference groups.

**2nd Assessment**								
Statistical Parameter	D1. Somatic Condition		D2. Psychological State		D3. Social Status		D4. Environment	
	Study Group	Reference Group	Study Group	Reference Group	Study Group	Reference Group	Study Group	Reference Group
Numbers	61	40	61	40	61	40	61	40
Average	65.6	59.6	66.7	62.1	58.3	47.7	60.0	58.1
Standard deviation	14.4	15.6	15.9	19.1	20.1	19.1	15.2	17.3
Minimum	25	0	29	4	8	0	34	0
25th percentile (bottom quartile)	57	54	54	53	42	33	50	50
50th percentile (median)	68	61	67	67	58	50	59	59
75th percentile (upper quartile)	75	71	75	75	67	60	72	72
Maximum	96	82	100	96	100	75	100	84
Test of normality of distribution Kolmogorov–Smirnov	NS (*p* > 0.20)	NS (*p* > 0.20)	NS (*p* > 0.20)	NS (*p* > 0.20)	NS (*p* > 0.20)	NS (*p* > 0.20)	NS (*p* > 0.20)	NS (*p* > 0.20)
Student’s *t*-test for unrelated samples	NS (*p* = 0.50)		NS (*p* = 0.19)		*p* < 0.0011		NS (*p* = 0.55)	

**Table 5 ijerph-20-04533-t005:** Descriptive statistics of individual changes in domain values (standardized to 0–100 points) of the WHOQOL-BREF scale in the study and reference groups, and the result of the significance test of these changes.

**Value Change = (2nd Assessment–1st Assessment)**								
Statistical Parameter	D1. Somatic Condition		D2. Psychological State		D3. Social Status		D4. Environment	
	Study Group	Reference Group	Study Group	Reference Group	Study Group	Reference Group	Study Group	Reference Group
Numbers	61	40	61	40	61	40	61	40
Average	3.7	0.4	−0.34	−0.52	−5.5	1.2	−1.6	1.3
Standard deviation	16.5	8.4	15.0	7.7	21.7	17.6	16.0	12.3
Minimum	−50	−18	−37	−25	−58	−25	−38	−19
25th percentile (bottom quartile)	−4	−4	−8	−4	−17	−8	−13	−6
50th percentile (median)	0	0	−4	0	−8	0	−3.1	0
75th percentile (upper quartile)	11	4	8.3	4	8	8	6	3
Maximum	46	21	38	21	42	50	41	41
Student’s *t*-test of the significance of group changes	NS (*p* = 0.09)	NS (*p* = 0.79)	NS (*p* = 0.86)	NS (*p* = 0.67)	NS (*p* = 0.06)	NS (*p* = 0.65)	NS (*p* = 0.43)	NS (*p* = 0.52)

**Table 6 ijerph-20-04533-t006:** ANCOVA and ANOVA test results of comparison of domain values (standardized to 0–100 points) of the WHOQOL-BREF scale for different quality factors among patients of the study group for two measurements (for ANCOVA test, predictor = age).

Study Group (n = 61)								
Quantitative Predictor = Age/Qualitative Factor	D1. Somatic Condition		D2. Psychological State		D3. Social Status		D4. Environment	
Measurement	1	2	1	2	1	2	1	2
Age	*p* < 0.0011	---	*p* < 0.0011	---	*p* < 0.0011	---	*p* = 0.05	---
Marital status	NS (*p* = 0.31)	NS (*p* = 0.59)	NS (*p* = 0.60)	NS (*p* = 0.90)	NS (*p* = 0.17)	NS (*p* = 0.70)	NS (*p* = 0.90)	NS (*p* = 0.43)
Age	*p* < 0.0011	---	*p* < 0.0011	---	*p* < 0.001	---	*p* < 0.0011	---
Residence	NS (*p* = 0.43)	NS (*p* = 0.43)	NS (*p* = 0.43)	NS (*p* = 0.82)	NS (*p* = 0.16)	NS (*p* = 0.86)	NS (*p* = 0.06)	NS (*p* = 0.28)
Age	*p* < 0.0011	---	*p* < 0.0011	---	*p* < 0.001	---	*p* = 0.02	---
Education	NS (*p* = 0.70)	NS (*p* = 0.28)	NS (*p* = 0.13)	NS (*p* = 0.11)	NS (*p* = 0.33)	NS (*p* = 0.80)	NS (*p* = 0.74)	NS (*p* = 0.60)
Age	*p* < 0.0011	---	*p* < 0.0011	---	*p* < 0.001	---	*p* = 0.04	---
Labor activity	*p* = 0.05	*p* < 0.00116	*p* = 0.05	*p* < 0.00112	NS (*p* = 0.83)	NS (*p* = 0.13)	NS (*p* = 0.32)	NS (*p* = 0.26)
Age	*p* < 0.00114	---	*p* < 0.0011	---	*p* < 0.0011	---	*p* = 0.03	---
Length of stay in the ward	NS (*p* = 0.70)	NS (*p* = 0.46)	NS (*p* = 0.13)	NS (*p* = 0.69)	NS (*p* = 0.87)	NS (*p* = 0.96)	NS (*p* = 0.93)	NS (*p* = 0.07)
Age	*p* < 0.0011	---	*p* < 0.0011	---	*p* < 0.0011	---	*p* = 0.04	---
CGI-S scale	*p* = 0.05	NS (*p* = 0.15)	NS (*p* = 0.52)	NS (*p* = 0.34)	NS (*p* = 0.16)	NS (*p* = 0.27)	NS (*p* = 0.17)	NS (*p* = 0.53)
Age	*p* < 0.0011	---	*p* < 0.0011	---	*p* < 0.001	---	*p* = 0.02	---
Type of antipsychotic drug	NS (*p* = 0.70)	NS (*p* = 0.85)	NS (*p* = 0.47)	NS (*p* = 0.86)	NS (*p* = 0.70)	NS (*p* = 0.08)	NS (*p* = 0.81)	NS (*p* = 0.45)
Age	*p* = 0.05	---	*p* < 0.0011	---	*p* < 0.0011	---	*p* = 0.05	---
Number of correct answers	*p* = 0.02	NS (*p* = 0.89)	NS (*p* = 0.26)	NS (*p* = 0.34)	NS (*p* = 0.17)	NS (*p* = 0.77)	NS (*p* = 0.11)	NS (*p* = 0.95)

**Table 7 ijerph-20-04533-t007:** Interdependence of ratings of changes, between 2nd and 1st assessments, in the number of correct answers to the knowledge test questions, and ratings of changes in the difference in domain scores of the WHOQOL-BREF scale for the results in the study group. The odds ratio (OR) of an increase in domain scores, with an increase in the number of correct answers, is given.

Domain	Assessing the Change in Domain Value	Evaluation of Changes in the Number of Correct Answers		Odds Ratio OR (95% Confidence Interval)	Significance Test
		Growth	Lack of Growth		
All patients of the study group with two assessments					
D1. Somatic condition	growth	20 (32.8%)	10 (16.4%)	1.26 (0.44;3.60)	NS (*p* = 0.66)
	lack of growth	19 (31.1%)	12 (19.7%)		
D2. Psychological state	growth	19 (31.1%)	6 (9.8%)	2.53 (0.82;7.83)	NS (*p* = 0.09)
	lack of growth	20 (32.8%)	16 (26.2%)		
D3. Social status	growth	12 (19.7%)	5 (8.2%)	1.51 (0.45;5.05)	NS (*p* = 0.36)
	lack of growth	27 (45.9%)	17 (27.9%)		
D4. Environment	growth	16 (26.2%)	9 (14.8%)	1.00 (0.34;2.90)	NS (*p* = 0.99)
	lack of growth	23 (37.7%)	13 (21.3%)		
Patients of the study group with less than 34 correct answers in 1st assessment					
D1. Somatic condition	growth	11 (36.7%)	6 (20.0%)	1.15 (0.26;5.11)	NS (*p* = 0.86)
	lack of growth	8 (26.7%)	5 (16.7%)		
D2. Psychological state	growth	11 (36.7%)	2 (6.7%)	6.19 (1.04;36.8)	*p* = 0.05
	lack of growth	8 (26.7%)	9 (30.0%)		
D3. Social Status	growth	6 (20.0%)	3 (10.0%)	1.23 (0.24;6.36)	NS (*p* = 0.80)
	lack of growth	13 (43.3%)	8 (26.7%)		
D4. Environment	growth	8 (26.7%)	5 (16.7%)	0.87 (0.20;3.90)	NS (*p* = 0.86)
	lack of growth	11 (36.7%)	6 (20.0%)		

**Table 8 ijerph-20-04533-t008:** Descriptive statistics of the number of correct answers in the knowledge test among patients of the study and reference group for the 1st and 2nd assessments.

Statistical Parameter	1st Assessment		2nd Assessment	
	Study Group	Reference Group	Study Group	Reference Group
Numbers	67	48	61	40
Average	32.9	31.2	34.3	30.5
Standard deviation	4.0	5.1	4.6	4.3
Minimum	19	19	17	22
25th percentile (bottom quartile)	30	29	32	28
50th percentile (median)	34	32	35	31
75th percentile (upper quartile)	36	35	37	33
Maximum	40	40	40	39
Test of normality of distribution Kolmogorov–Smirnov	*p* < 0.01	NS (*p* > 0.20)	*p* < 0.0015	NS (*p* > 0.20)
Mann–Whitney U test of comparisons between groups	NS (*p* = 0.09)		*p* < 0.0011	
Wilcoxon paired-order test for group changes	Study group: *p* < 0.0011 Reference group: NS (*p* = 0.59)			

## Data Availability

Data supporting reported results are available on demand from J.F.

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
