# Peer review of "The Impact of Health Education on the Quality of Life of Patients Hospitalized in Forensic Psychiatry Wards"

_ijerph, 2023, doi:10.3390/ijerph20054533_

Round 1
Reviewer 1 Report
Lines 15-18. This sentence needs to be restructured. It does not make sense as it is written.
Lines 71-73 is one sentence but is presented as a paragraph. A paragraph should be more than one sentence.
Lines 77-79 needs rewording.
Lines 83-86 is not a complete sentence. This needs to be reworded.
Line 101 Why did this not require approval from a Review Board? Forensic psychiatric patients are a vulnerable population.
General punctuation is confusing throughout the manuscript. There are errors in subject/verb regarding singular vs plural. Misused future, present, and past tense verbs. I suggest authors have someone proficient in English to assist in editing the paper.
Punctuation in tables (especially decimals) is confusing. Tables in general need work.
Unable to determine references in some cases due to typing errors??
Author Response
Point 1: Lines 15-18. This sentence needs to be restructured. It does not make sense as it is written.
Response 1: The introductory part in the Abstract was rewritten to the present form: “An original health education program developed for a group of patients of forensic psychiatry wards was the basis for conducting a study on the impact of educational influences on the quality of life of patients long-term isolated from their natural environment. The main aim of the study was to answer the question: Does health education affect the quality of life of patients in forensic psychiatry wards and is educational activity effective?”
[lines 15-19]
Point 2: Lines 71-73 is one sentence but is presented as a paragraph. A paragraph should be more than one sentence.
Response 2: The sentence was rephrased to: Scientific research clearly shows that pharmacological treatment in combination with psychosocial interactions is an important element of therapeutic programs aimed at helping people with schizophrenia recover [6, 7].” And included into the next paragraph [lines 70-72].
Point 3: Lines 77-79 needs rewording.
Response 3: The part of the paragraph referring to the role of the nurse in the forensic psychiatry word was rewritten: “The nurse of the forensic psychiatry ward is the person closest to the patient, thanks to which she is an unquestionable source of information about the patient's changing physical and mental condition. The knowledge about the patient obtained by nurses is a reliable foundation for both treatment and rehabilitation because patients of forensic psychiatry wards stay there long after their mental illness has long stopped to be a leading problem.” [lines: 75-80]
Point 4: Lines 83-86 is not a complete sentence. This needs to be reworded.
Response 4: The statement was rewritten: “The health education program for mentally ill offenders developed by the first author was the starting point for examining the impact of educational program on the quality of life of patients long-term isolated in a forensic psychiatry ward.” [lines: 81-83]
Point 5: Line 101 Why did this not require approval from a Review Board? Forensic psychiatric patients are a vulnerable population.
Response 5: The study was approved by the bioethics committee to be conducted. The Commission waved only the obligation to collect patients' consent to participate in the study. A supplement has been added to the paragraph: “Institutional Review Board Statement: The study was conducted in accordance with the Declaration of Helsinki, and approved by the Ethics Committee of Silesian Medical University in Katowice (protocol code PCN/0022/KB/133/20 from 13/08/2020)” [lines 442-444].
Point 6: General punctuation is confusing throughout the manuscript. There are errors in subject/verb regarding singular vs plural. Misused future, present, and past tense verbs. I suggest authors have someone proficient in English to assist in editing the paper.
Punctuation in tables (especially decimals) is confusing. Tables in general need work.
Unable to determine references in some cases due to typing errors??
Response 6: We would like to thank the reviewer for his patience when reading the manuscript and detailed analysis of the text. All the syntax and grammatical errors were corrected. All the body of text was rewritten and then corrected by the English editor. Punctuation in tables was adjusted to the English language.

Reviewer 2 Report
The study reports on the investigation of the relationship between health education programs and Quality of Life (QOL) among patients in the psychiatric ward of a forensic medicine classroom. The study utilized the WHOQOL-BREF standardized QOL assessment scale, measuring QOL both qualitatively and quantitatively. This study is significant in exploring the potential benefits of health education for enhancing QOL among patients in the psychiatric ward of the forensic medicine classroom, which requires a comprehensive approach.
However, the following concerns exist:
Major
(i) More information is needed on the reason why approval from an ethics committee was not required. In general, ethics committee approval is necessary when conducting research involving the health information of individuals. If the ethics committee has determined that approval is not necessary, it would be appropriate to indicate this in the study in a transparent and impartial manner. It would also be desirable to specify whether informed consent was obtained from the patients. If the reason for not requiring ethics committee approval is reasonable, it would be appropriate to specify that the study was performed based on the Helsinki Declaration.
(ii) The content of health education is unclear. Is it a comprehensive program or something developed independently? Was the health education provided only once? Is the intervention aimed at a group or individual to an individual? Please provide clarification on how the health education intervention was carried out. It does not describe what health education interventions were used, so other psychiatric wards cannot take full advantage of this paper.
(iii) The basic statistical information about the case-control group is unclear. As shown in Table 5, variables such as marital status, residence, education, labor activity, length of stay in the ward, CGI-S scale, type of neuroleptic used, and the number of correct answers to the knowledge test should be shown as basic statistics of the case and control groups, and differences between the groups should be discussed.
Minor
(i) Please translate Polish in Table 6 into English.
(ii) Please adjust the p-value in Table 7 to the number of significant digits specified by the editor.
Author Response

(The authors gave the same response as above.)

Round 2
Reviewer 2 Report
Thank you for addressing each of my previous comments appropriately. I would like to request a minor correction below.
1) For each table, please correct the notation "p=0.00" and "p<0.00" to "p<0.001," as this is the more commonly used notation.
Author Response
Point 1. For each table, please correct the notation "p=0.00" and "p<0.00" to "p<0.001," as this is the more commonly used notation.
Response 1: It was corrected according to the reviewer’s comment.
